# DualCoOp: Fast Adaptation to Multi-Label Recognition with Limited Annotations

**Ximeng Sun**[1]    **Ping Hu**[1]    **Kate Saenko**[1,2]
[1]Boston University, [2]MIT-IBM Watson AI Lab, IBM Research
{sunxm, pinghu, saenko}@bu.edu

## Abstract

Solving multi-label recognition (MLR) for images in the low-label regime is a challenging task that has many real-world applications. Recent work learns an alignment between textual and visual spaces to compensate for insufficient image labels, but loses accuracy because of the limited amount of available MLR annotations. In this work, we utilize the strong alignment of textual and visual features pretrained with millions of auxiliary image-text pairs and propose *Dual Context Optimization* (`DualCoOp`) as a unified framework for partial-label MLR and zero-shot MLR. `DualCoOp` encodes positive and negative contexts with class names as part of the linguistic input (i.e. prompts). Since `DualCoOp` only introduces a very light learnable overhead upon the pretrained vision-language framework, it can quickly adapt to multi-label recognition tasks that have limited annotations and even unseen classes. Experiments on standard multi-label recognition benchmarks across two challenging low-label settings demonstrate the advantages of our approach over state-of-the-art methods. Project page: `https://cs-people.bu.edu/sunxm/DualCoOp/project.html`

## 1   Introduction

Image recognition has become a very popular and successful research area in recent years, due to the development of large-scale datasets [12, 28] and advanced model architectures [14, 19, 38, 54]. However, the majority of image recognition approaches have focused on single-label prediction, which ignores the intrinsic multi-label nature of images. Unlike single-label recognition [14, 19, 38, 54], multi-label image recognition aims to recognize all semantic labels present in an image [10, 11, 35, 37, 52, 66, 61], providing a more comprehensive understanding and benefiting applications like image retrieval, video analysis, and recommendation systems.

Multi-label recognition typically deals with images of complex scenes and diverse objects. Collecting multi-label annotations becomes difficult to scale up, for two reasons: (i) annotating images with the full semantic label set is laborious and (ii) samples of particular categories can be hard to find. The first challenge can be addressed by multi-label recognition with *partial labels*, where merely some of the categories are annotated for each training image. Recent works proposed solutions to partial-label MLR based on semi-supervised learning [24, 41], normalized training objectives [15], or label correlations [8, 21, 46]. The second setting involves *zero-shot* MLR, where novel unseen categories are recognized by transferring knowledge from seen categories, with solutions like principal image features [3, 69], knowledge graphs [29], and attention mechanisms [22, 44]. Despite significant progress on the two settings, existing approaches are not designed to handle both at once. We propose to unify these settings as *limited-annotation* MLR and design a solution that can handle practical scenarios with either partial or missing labels.

Successful solutions to the above problems transfer knowledge from fully-annotated categories to partially-labeled and novel categories by learning an alignment between images and category

36th Conference on Neural Information Processing Systems (NeurIPS 2022).

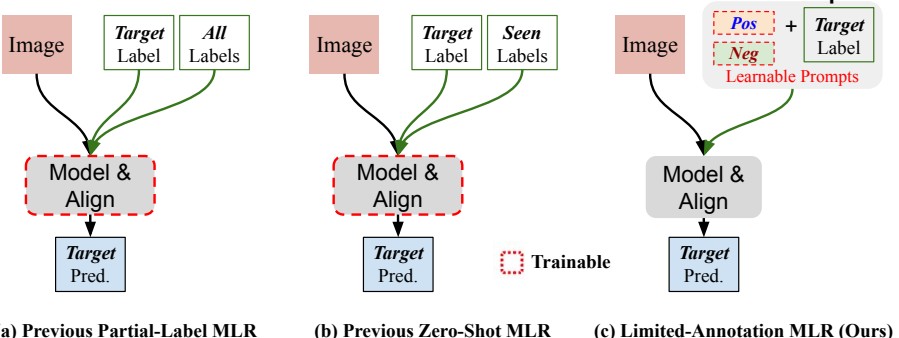

(a) Previous Partial-Label MLR     (b) Previous Zero-Shot MLR     (c) Limited-Annotation MLR (Ours)

Figure 1: **A conceptual comparison of previous multi-label recognition (MLR) methods and our approach**. In Partial-Label MLR (a) and Zero-Shot MLR (b), previous works learn to model and align the visual and textual inputs as well as explore the correlation between the target label with all/seen labels depending on the limited semantic annotations available on the dataset, which leads to sub-optimal performance and complex model architectures. In contrast, we propose a unified framework to tackle both limited-annotation tasks (c). We rely on the modeling and alignment of visual and textual inputs contained a large-scale pretrained vision-language model, and only learn a pair of positive and negative prompts to this model.

names [8, 46, 69]. Recently, vision-language pretraining models are bridging the visual-textual gap via large-scale pretraining, e.g., CLIP [47] is trained with 400 million image-text pairs. In this work, we draw inspiration from the recent success of prompt learning for such models [20, 25, 40, 48, 70, 72]. Prompt learning provides a convenient way to transfer pretrained vision-language models to other tasks. It designs additional templated or learnable prompt tokens for textual input to "inform" the model about downstream tasks and avoids finetuning the entire model which can be inefficient and data-hungry. By doing so, recent works like CoOp [71] have demonstrated CLIP's remarkable generalisation to various zero-shot image tasks [20, 47, 72]. However, these methods mainly focus on matching each image with a single label, hence they are not able to handle the multi-label setting.

To adapt the knowledge learned in CLIP to multi-label image recognition, we propose the `DualCoOp` framework. As shown in Fig. 1 (c), `DualCoOp` learns a pair of differentiable prompts to provide positive and negative contexts for the target class. Instead of using hand-crafted thresholding to determine positive labels [51], the dual prompts naturally result in a positive and a negative classifier, so the existence of the target class in the image can be easily decided by comparing their scores. Unlike prior models, shown in Fig. 1 (a)(b), we avoid fine-tuning the full vision-language model and only learn the prompts, which are much smaller compared to the entire model. Therefore, our simple framework achieves much higher efficiency when adapting to different datasets. Additionally, we modify the attention mechanism of CLIP to better model spatial information in images, improving its ability to recognize multiple objects in MLR. With these design choices, we achieve a unified framework for addressing the general challenges of multi-label recognition with limited annotations.

We summarize our contributions as follows:

- We propose `DualCoOp` to quickly adapt powerful vision-language models to solve multi-label recognition tasks using limited annotations.
- We propose dual (positive and negative) prompts to drastically reduce the number of learnable parameters, and improve the spatial modeling of the visual encoder to better distinguish multiple objects.
- We conduct extensive experiments on partial-label MLR (on MS-COCO [34] and VOC2007 [16]) and zero-shot MLR (on MS-COCO and NUS-WIDE [11]). Notably, `DualCoOp` improves mAP by 6.8% with 10% of labels on VOC2007, and F1-score at Top-3 Prediction by 10.8% for zero-shot MLR on NUS-WIDE.

## 2   Related Works

**Multi-Label Recognition with Limited Annotations.** Multi-label image recognition has drawn increasing attention in past years. One straightforward solution to this problem is to individually learn a binary classifier for each category [36, 43, 57], which however does not consider correlations

among labels. Hence, recent works have focused on incorporating semantic dependencies among labels via graph neural networks [10, 11, 61] or RNN/LSTM [35, 59, 62, 66]. Some work also considers the spatial distribution of labels in the image, and exploits object proposals [33, 37, 60] or attention mechanism [52, 62, 73] as a regularization to rectify the prediction. However, despite achieving significant progress, these methods require a large-scale and complete annotated dataset to train models [27, 34]. This limits their application to more practical scenarios where data is partially annotated for training [4, 6, 41, 55, 56, 64] and unseen (zero-shot) categories may appear during testing [7, 18, 29, 42, 69].

With partially labeled data, where merely some labels of each sample are known, Mahajan *et al.* [41] and Joulin *et al.* [24] attempt to use web supervision to automatically generate the pseudo labels, which unfortunately leads to poor performance as the web supervision is noisy and incomplete [67]. To avoid external noise, Durand *et al.* [15] exploit the proportion of annotated samples for different labels and propose a normalized BCE loss to train models based on the given partial labels. More recent works explicitly transfer information from known labels to complement unknown labels by utilizing category-specific feature blending [46] or label co-occurrences [8] at both instance-level and prototype-level.

Unlike partial annotation of the same label set for training and testing, zero-shot multi-label image recognition needs to handle novel categories during testing, hence inspiring a different route based on a joint visual-label embedding space [7, 22, 42, 44, 69]. Zhang *et al.* [69] propose to find a principal direction that ranks related labels first in the joint embedding space optimized via a tailored zero-shot ranking loss. Cohen *et al.* [3] further improve the idea by learning multiple principal vectors to support the semantic diversity. Huynh *et al.* [22] consider the spatial regularization and propose a shared multi-attention model and obviate the need for explicit region proposals [50]. Narayan *et al.* [44] propose to enhance the region-based features so as to minimize inter-class feature entanglement.

Though significant progress has been made in each of the directions, existing methods still require a lot of MLR data and complex architectures/losses. Our approach reduces the need for hard-to-get MLR data by pretraining on unsupervised text-image pairs. While it may seem unfair to compare existing MLR methods with ones based on such pretraining, we point out that the pretraining data is unsupervised and thus easier to obtain. We also provide experiments comparing `DualCoOp` to baselines using the same pretraining. Importantly, previous methods are designed for only one task, hence have limitations in practical applications. In contrast, our proposed framework can be easily adapted with small data and can address both partial and zero-shot tasks at the same time.

**Prompt Learning for Vision-Language Models.** Vision-Language Models [23, 47] based on contrastive learning have demonstrated impressive ability to learn generic visual representations. As a milestone, CLIP [47] is trained with 400 million curated image-text pairs, and shows remarkable transfer capability for over 30 classification datasets. With such powerful vision-language models, several follow-ups [17, 63, 65, 68] have been proposed to explore the training strategies for training downstream classification tasks. Instead of fine-tuning the entire model [13, 19], which may damage the learned representation space, recent approaches adopt the prompt-based paradigm that formalizes NLP tasks as masked language modeling (prompt templates) [30, 32, 53]. Zhou *et al.* [71] propose to tune prompts for downstream classification tasks, and further introduce input-conditional prompts for better generalization ability [72]. Lu *et al.* [39] learn the distribution of diverse prompts to handle the varying visual representations. Huang *et al.* [20] generate pseudo labels for images to learn prompts in an unsupervised way. Though achieving promising improvements for downstream tasks, these methods address the multi-class zero-shot image recognition, assuming each image has one label, hence lacking the ability to handle the multi-label setting. In this paper, we present a novel framework to efficiently transfer VLMs to address multi-label image recognition with limited annotations.

## 3  Method

**Problem Definition.** We formally define multi-label recognition with limited annotations as follows: Consider $M$ as the set of categories which describe objects or attributes in images. Given a training image $I$, the existence of a category $m \in M$ can be positive, negative or unknown, corresponding to the label $y_m = 1, -1$ or $0$ respectively. During inference, we predict each label of interest for an input image.

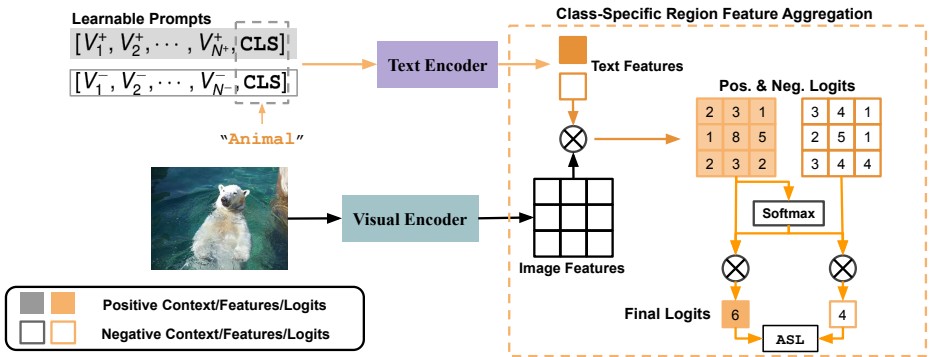

Figure 2: **Illustration of our proposed approach.** `DualCoOp` learns a pair of positive and negative prompts to quickly adapt powerful pretrained vision-text encoders to the MLR task. For each class, two prompts generate two contrastive (positive and negative) textual embeddings as the input to the text encoder. Furthermore, we propose *Class-Specific Region Feature Aggregation* to first project each region's feature to the textual space and then aggregate the spatial logits by the magnitude of class-specific semantic responses. During training, we apply the ASL loss [51] to optimize learnable prompts while keeping other network components frozen. During inference, we compare the positive and negative logits to make a prediction for each class.

Many existing MLR problems fit into this broad definition. In this paper, we consider the settings with partial or missing labels: (1) **Partial-label MLR** [8, 15, 46], in which only a subset of labels are known ($+1$ or $-1$) for each training image and we are interested in predicting all existing labels during inference. (2) **Zero-shot MLR** [3, 22, 49], in which each label is either known (seen) or unknown (unseen) for *all* images during training and we are interested in predicting either all labels or only unknown (unseen) labels during inference. In this paper, we propose a unified setting that includes both scenarios, which we call **limited-annotation MLR**.

**Approach Overview.** To compensate for insufficient or missing image labels, it is important to learn how the meanings of category names are related to each other, so that we can transfer knowledge between related categories. This is usually done by learning an alignment between the visual and textual spaces. However, our dataset is too limited to learn a broad and generalizable mapping. We propose to instead leverage the strong alignment of visual and textual feature spaces learned by large-scale vision-language pretraining (CLIP [48]) with a light-weight learnable overhead which quickly adapts to the MLR task with limited semantic annotations. Figure 2 provides an overview of our proposed approach. `DualCoOp` learns a pair of "prompt" contexts in the form of two learnable sequences of word vectors, to provide positive and negative contexts of a given category name $m$. This generates positive and negative textual features $(F_t^m)_+$ and $(F_t^m)_-$ that are fed into the pretrained text encoder. Furthermore, to better recognize multiple objects, which can be located at different locations in the image, the spatial aggregation step is modified. We first compute the similarity score of each projected visual feature $F_v^i$ at location $i$ with $(F_t^m)_+/(F_t^m)_-$ to obtain prediction logits over regions. For each class, we perform aggregation of all spatial logits, in which the weight for each logit is determined by its relative magnitude. We call this *Class-Specific Region Feature Aggregation*. During training, we optimize the learnable prompts via the ASL loss [51] while keeping all other network components frozen. During inference, we directly compare the final positive and negative logits to make a prediction for each label $y_m$.

**Dual Learnable Prompts.** Instead of learning a single prompt for a class [71], we propose Dual Context Optimization (`DualCoOp`) which learns two contrastive prompts' contexts for each class. The learnable part in dual prompts carries positive and negative contextual surroundings individually and can be optimized end-to-end from data via binary classification loss. Specifically, we define the pair of prompts given to the text encoder as follows:

$$\text{Prompt}^+ = \left[V_1^+, V_2^+, \cdots, V_{N^+}^+, \text{CLS}\right], \tag{1}$$

$$\text{Prompt}^- = \left[V_1^-, V_2^-, \cdots, V_{N^-}^-, \text{CLS}\right] \tag{2}$$

where each $V$ is a learnable word embedding vector (*e.g.* with dimension 512 in CLIP [48]) and `CLS` is the given category name. $N^+$ and $N^-$ are the numbers of word tokens learned in the positive and

negative prompts respectively. For simplicity, we set $N^+ = N^-$ in our experiments. We learn a pair of positive and negative prompts for each class (i.e. class-specific prompt pair) when solving MLR with partial labels, and learn a pair of prompts shared for all classes in zero-shot MLR. With a pair of prompts, we compute the binary classification output $p$ with the following form:

$$p = \frac{\exp(< A(E_v(I)), E_t(\text{Prompt}^+) > /\tau)}{\exp(< A(E_v(I)), E_t(\text{Prompt}^+) > /\tau) + \exp(< A(E_v(I)), E_t(\text{Prompt}^-) > /\tau)}, \quad (3)$$

where $< \cdot, \cdot >$ represents cosine similarity and $p$ is the predicted probability for a given (image, label) pair as positive example. $E_v(\cdot)$ and $E_t(\cdot)$ are the visual and textual encoders from the vision-language pretraining. $A(\cdot)$ is our new aggregation function to adaptively reduce the spatial dimension of visual features for each class, which will be discussed next.

**Class-Specific Region Feature Aggregation.** In multi-label image recognition, it is common that multiple objects appear in different regions of the image. Pooling to produce a single image-level feature vector for all classes gives sub-optimal performance since spatial information is reduced and different objects are mixed. In this work, we reformulate the last multi-headed attention pooling layer of the visual encoder in CLIP [48] and apply class-specific pooling to adaptively aggregate region features in the multi-label setting. The original attention pooling layer in CLIP pools the visual feature map first, and then projects the global feature vector into text space as follows:

$$\text{AttnPool}(x) = \text{Proj}_{v \to t}(\sum_i \text{softmax}(\frac{q(\bar{x})k(x_i)^T}{C}) \cdot v(x_i))$$

$$= \sum_i \text{softmax}(\frac{q(\bar{x})k(x_i)^T}{C}) \cdot \text{Proj}_{v \to t}(v(x_i)) = \text{Pool}(\text{Proj}_{v \to t}(v(x_i))),$$

where $q$, $v$ and $k$ are independent linear embedding layers and $x = E_v(I)$ is the output feature map of the visual encoder. By removing the pooling operation, we can project the visual feature $x_i$ of each region $i$ to the textual space [70]:

$$F_v^i = \text{Proj}_{v \to t}(v(x_i)). \quad (4)$$

For each region $i$ and each class $m$, we compute cosine similarity between $F_v^i$ and $(F_t^m)^+ = E_t(\text{Prompt}^+)$ as $S_{i,m}^+ = < F_v^i, (F_t^m)^+ >$, and compute $S_{i,m}^-$ in the same way. In order to make a single prediction for the whole image, we aggregate $S_{i,m}^+$ and $S_{i,m}^-$ into $S_m^+$ and $S_m^-$ according to the magnitude of $S_{i,m}^+$, i.e.:

$$S_m^+ = A(S_{i,m}^+) = \sum_i \left(\text{softmax}(S_{i,m}^+) \cdot S_{i,m}^+\right), \quad (5)$$

$$S_m^- = A(S_{i,m}^-) = \sum_i \left(\text{softmax}(S_{i,m}^+) \cdot S_{i,m}^-\right). \quad (6)$$

Notably, we do not introduce any new parameters in our re-formulation of the spatial aggregation function. All parameters used to project visual features to the textual space are inherited from the original multi-headed attention pooling layer in CLIP.

**Optimization**. We apply the Asymmetric Loss (ASL) [51] to handle the inherent positive-negative imbalance in the optimization of multi-label recognition. Specially, we compute losses for a positive (image, label) pair $\mathcal{L}_+$ and a negative (image, label) pair $\mathcal{L}_-$ as follows:

$$\mathcal{L}_+ = (1 - p)^{\gamma_+} \log(p), \quad (7)$$
$$\mathcal{L}_- = (p_c)^{\gamma_-} \log(1 - p_c), \quad (8)$$

where $p_c = \max(p - c, 0)$ is the probability for negative examples shifted by hard thresholding via the margin $c$. We set the hyper-parameters $\gamma_- \geq \gamma_+$, so that ASL down-weighs and hard-thresholds easy negative samples. The pair of learnable prompts are updated by back-propagating ASL through the frozen text encoder.

# 4 Experiments

## 4.1 Multi-Label Recognition with Partial Labels

**Datasets.** We conduct experiments on MS-COCO [34], VOC2007 [16] and BigEarth [5] to evaluate multi-label recognition with partial labels. MS-COCO [34] contains 80 common object categories

Table 1: **Multi-label Recognition on MS-COCO and VOC2007 with partial labels.** DualCoOp achieves the best performance over all SOTA methods. * indicates previous models using weights pretrained by CLIP [47]

| Methods | #P | 10% | 20% | 30% | 40% | 50% | 60% | 70% | 80% | 90% | Avg. |
|---|---|---|---|---|---|---|---|---|---|---|---|
| | | | | | MS-COCO [34] | | | | | | |
| SSGRL [9] | 64.7M | 62.5 | 70.5 | 73.2 | 74.5 | 76.3 | 76.5 | 77.1 | 77.9 | 78.4 | 74.1 |
| GCN-ML [10] | 44.9M | 63.8 | 70.9 | 72.8 | 74.0 | 76.7 | 77.1 | 77.3 | 78.3 | 78.6 | 74.4 |
| KGGR [7] | ≥ 25M | 66.6 | 71.4 | 73.8 | 76.7 | 77.5 | 77.9 | 78.4 | 78.7 | 79.1 | 75.6 |
| Curriculum labeling [15] | ≥ 38M | 26.7 | 31.8 | 51.5 | 65.4 | 70.0 | 71.9 | 74.0 | 77.4 | 78.0 | 60.7 |
| Patial BCE [15] | ≥ 38M | 61.6 | 70.5 | 74.1 | 76.3 | 77.2 | 77.7 | 78.2 | 78.4 | 78.5 | 74.7 |
| SST [8] | 33.5M | 68.1 | 73.5 | 75.9 | 77.3 | 78.1 | 78.9 | 79.2 | 79.6 | 79.9 | 76.7 |
| SST* | 33.5M | 69.1 | 78.5 | 79.3 | 79.9 | 80.1 | 80.5 | 81.1 | 80.7 | 80.7 | 78.9 |
| SARB [46] | 29.6M | 71.2 | 75.0 | 77.1 | 78.3 | 78.9 | 79.6 | 79.8 | 80.5 | 80.5 | 77.9 |
| SARB* | 29.6M | 75.5 | 78.5 | 79.0 | 79.5 | 80.4 | 80.2 | 80.8 | 80.6 | 80.8 | 79.4 |
| DualCoOp (ours) | **1.3M** | **78.7** | **80.9** | **81.7** | **82.0** | **82.5** | **82.7** | **82.8** | **83.0** | **83.1** | **81.9** |
| | | | | | PASCAL VOC 2007 [16] | | | | | | |
| SSGRL [9] | 66.6M | 77.7 | 87.6 | 89.9 | 90.7 | 91.4 | 91.8 | 91.9 | 92.2 | 92.2 | 89.5 |
| GCN-ML [10] | 44.9M | 74.5 | 87.4 | 89.7 | 90.7 | 91.0 | 91.3 | 91.5 | 91.8 | 92.0 | 88.9 |
| KGGR [7] | ≥ 25M | 81.3 | 88.1 | 89.9 | 90.4 | 91.2 | 91.3 | 91.5 | 91.6 | 91.8 | 89.7 |
| Curriculum labeling [15] | ≥ 38M | 44.7 | 76.8 | 88.6 | 90.2 | 90.7 | 91.1 | 91.6 | 91.7 | 91.9 | 84.1 |
| Patial BCE [15] | ≥ 38M | 80.7 | 88.4 | 89.9 | 90.7 | 91.2 | 91.8 | 92.3 | 92.4 | 92.5 | 90.0 |
| SST [8] | 32.4M | 81.5 | 89.0 | 90.3 | 91.0 | 91.6 | 92.0 | 92.5 | 92.6 | 92.7 | 90.4 |
| SARB [46] | 29.6M | 83.5 | 88.6 | 90.7 | 91.4 | 91.9 | 92.2 | 92.6 | 92.8 | 92.9 | 90.7 |
| DualCoOp (ours) | **0.3M** | **90.3** | **92.2** | **92.8** | **93.3** | **93.6** | **93.9** | **94.0** | **94.1** | **94.2** | **93.2** |

and we use the official `train2014` (82K images) and `val2014` (40K images) splits for training and test. VOC2007 [16] contains 20 object categories and we use the official `trainval` (5K images) and `test` (5K images) splits for training and test. Furthermore, since CLIP pretraining data is not publicly available and it is plausible that CLIP pretraining data covers many coarse and fine-grained visual domains since it performs well in the zero-shot evaluation for many downstream tasks, we also experiment on a Remote Sensing Image dataset BigEarth [5], whose domain is far from the domains of the datasets in the mainstream papers (i.e. PASCAL VOC, MS-COCO, and NUS-WIDE). To create the training set with partial labels, we randomly mask out labels from the fully annotated training set[1] and use the remaining labels for training by following standard practice [8, 15, 46]. In this work, we vary the proportion of kept labels from 10% to 90% [8, 46].

**Evaluation.** On all datasets, we follow [8, 15, 46] to report the mean average precision (mAP) for each proportion of labels available for optimization (from 10% to 90%) and its average value for all proportions. We count the learnable parameters (#P) of each baseline and DualCoOp to measure the complexity of optimization[2]. We also report the per-class and the average overall precision (CP and OP), recall (CR and OR), and F1 (CF1 and OF1) of DualCoOp under different proportions of labels for training in the supplementary material due to the page limit.

**Implementation.** We adopt ResNet-101 [19] as the visual encoder in all baselines and DualCoOp for input resolution 448×448, and use the same Transformer [48, 58] in CLIP [47] as the text encoder. The visual and text encoders are initialized from the CLIP pretrained model and kept frozen during optimization. For each class/label, we learn two independent context vectors with 16 context tokens ($N = 16$) following [71], which is the only learnable part in DualCoOp. We use the SGD optimizer with an initial rate of 0.002 which is decayed by the cosine annealing rule. We train context vectors for 50 epochs with a batch-size 32/8/32 for MS-COCO/VOC2007/BigEarth, respectively. For ASL loss, we choose $\gamma_+ = 1$, $\gamma_- = 2$ and $c = 0.05$ via validation. Training is done with one RTX A6000.

**Baselines.** To evaluate the effectiveness of DualCoOp, we compare with the following baselines: (1).SSGRL [9], GCN-ML [10] and KGGR [7] adopt graph neural networks to model label dependencies. We follow [8] to report their performance in the partial-label setting. (2). Curriculum labeling [15] and SST [8] generate pseudo labels for unknown labels. (3). Partial BCE [15] uses a normalized BCE loss to better exploit partial labels. (4). SARB [46] blends category-specific representation across different images to transfer information of known labels to complement unknown labels.

---

[1]The difference in performance is within 1.0% of independent runs.

[2]For baselines without public released implementation, we only measure the major part of the learnable parameters based on description in their papers. (indicated as #P ≥ [a value] in Table 1-5)

Table 2: **Comparison between SARB and DualCoOp. Both use parameters pretrained by CLIP [47].**

| Method | 10% | 30% | 50% | 70% | 90% | Avg. |
|---|---|---|---|---|---|---|
| SARB* | 77.2 | 83.6 | 87.4 | 91.6 | 93.5 | 86.7 |
| DualCoOp | **88.2** | **93.1** | **93.7** | **94.2** | **94.7** | **92.8** |

Table 3: **Computations Cost Comparison between SARB [46] and DualCoOp.** * indicates previous models using weights pretrained by CLIP [47]

| Methods | Training Latency (ms/img) | Training Memory (GB/img) | Testing Latency (ms/img) | Testing Memory (GB/img) | Avg. mAP (%) |
|---|---|---|---|---|---|
| SARB* | 4.7 | 0.21 | 4.0 | 0.13 | 79.4 |
| DualCoOp | 5.3 | 0.22 | 4.0 | 0.06 | 81.9 |

**Results.** Table 1 shows the comparison of mAP between DualCoOp and all baselines optimized with 10% to 90% of labels. For the two most recent works (SST [8] and SARB [46]), we further substitute the ImageNet pretrained weights [19] with the CLIP pretrained weights [47] when initializing of their visual encoders, which results in SST* and SARB* in Table 1. Since we learn class-specific prompts, DualCoOp on MS-COCO adopts more learnable parameters than VOC2007. Our proposed DualCoOp achieves the best performance across all proportions of labels available during the training with the smallest learnable overhead (1.3M vs. 29.6M in SARB* on MS-COCO and 0.3M vs. 29.6M in SARB* on VOC2007). Notably, DualCoOp yields a great improvement over the second-best method, 3.2% on MS-COCO and 6.8% on VOC2007, especially when only providing 10% of labels during the training. This indicates that DualCoOp can quickly adapt to the multi-label recognition task with a few labels. On BigEarth, we compare DualCoOp with the strong baseline SARB. Table 2 shows that DualCoOp consistently improves over SARB by 11.0% when only providing 10% of labels and 6.1% for average performance which proves DualCoOp boosts the performance in various visual domains by taking advantage of the powerful vision-language pretraining.

**Full Label Training.** On MS-COCO, we use 100% of training labels and finetune the visual encoder, DualCoOp achieves 85.2% mAP, outperforming previous SOTA approaches like ASL [51] (85.0% mAP) and CSRA [74] (83.5% mAP) with the same ResNet-101 backbone. Without finetuning the visual encoder, we notice that our performance drops to 83.2% mAP, which is still comparable to the performance of ASL and CSRA. Such a performance drop may be due to the noise in the image-text pairs used for pre-training the CLIP model and the misalignment of the objectives between the pre-training task and MLR task. This also shows that it is non-trivial to address the MLR task with the pretrained CLIP model, even though it is pre-trained with larger-scale data.

**Computational Cost.** We compare the computational cost between DualCoOp and SARB [46] in terms of training/testing latency and memory (see Table 3) using the same device (one Nividia A100 GPU). For the current multi-label recognition task, the categories are pre-set before inference, (i.e. we already know which class we would like to consider during inference.) In this case, we compute the text features for each class from the learned prompts and the class name ahead of the inference. Then we use the pre-computed text features to predict each image during the test. Since the text features are pre-computed (very light-weighted computing overhead), the text encoder is not executed during inference. For inference, the latency time and memory consumption of DualCoOp are the same as other baselines when using the same backbone for the image encoder. During training, CLIP-based methods slightly raise latency time and memory consumption since image and text encoders are both executed during the forward, and only prompts are updated in DualCoOp.

### 4.2 Zero-shot Multi-Label Recognition

**Datasets.** Following [3, 22], we conduct experiments on MS-COCO [34] and NUS-WIDE [11] to perform zero-shot multi-label recognition. On MS-COCO, we follow [2, 3] to split the dataset into 48 seen classes and 17 unseen classes. NUS-WIDE [11] dataset includes 270K images. Following [3, 22] we use 81 human-annotated categories as unseen classes and an additional set of 925 labels obtained from Flickr tags as seen classes.

**Evaluation.** We follow [3] and report precision, recall, and F1 score at Top-3 predictions in each image on MS-COCO. We also follow [3, 22] to report mAP over all categories as well as precision, recall, and F1 score at Top-3 and Top-5 predictions in each image on NUS-WIDE. We evaluate all

Table 4: **Zero-Shot Multi-Label Recognition on MS-COCO[34]**. DualCoOp achieves the best F1 score in both ZSL and GZSL settings.

| Methods | #P | ZSL | | | GZSL | | |
|---|---|---|---|---|---|---|---|
| | | **P** | **R** | **F1** | **P** | **R** | **F1** |
| CONSE [45] | - | 11.4 | 28.3 | 16.2 | 23.8 | 28.8 | 26.1 |
| Fast0Tag [69] | 0.61M | 24.7 | 61.4 | 25.3 | 38.5 | 46.5 | 42.1 |
| Deep0Tag [49] | ≥ 23M | 26.5 | 65.9 | 37.8 | 43.2 | 52.2 | 47.3 |
| SDL (M=2) [3] | 30.6M | 26.3 | 65.3 | 37.5 | **59.0** | 60.8 | 59.9 |
| DualCoOp (ours) | **0.02M** | **35.3** | **87.6** | **50.3** | 58.4 | **68.1** | **62.9** |

Table 5: **Zero-Shot Multi-label Recognition on NUS-WIDE [11]**. DualCoOp achieves the best F1 score over all SOTA methods at Top-3/Top-5 predictions in both ZSL and GZSL settings.

| Methods | #P | Top-3 | | | Top-5 | | | mAP |
|---|---|---|---|---|---|---|---|---|
| | | **P** | **R** | **F1** | **P** | **R** | **F1** | |
| *Zero-Shot Learning (ZSL)* | | | | | | | | |
| CONSE [45] | - | 17.5 | 28.0 | 21.6 | 13.9 | 37.0 | 20.2 | 9.4 |
| LabelEM [1] | - | 15.6 | 25.0 | 19.2 | 13.4 | 35.7 | 19.5 | 7.1 |
| Fast0Tag [69] | 0.61M | 22.6 | 36.2 | 27.8 | 18.2 | 48.4 | 26.4 | 15.1 |
| One Attention per Label [26] | ≥ 12.8M | 20.9 | 33.5 | 25.8 | 16.2 | 43.2 | 23.6 | 10.4 |
| LESA (M=10) [22] | ≥ 0.45M | 25.7 | 41.1 | 31.6 | 19.7 | 52.5 | 28.7 | 19.4 |
| BiAM [44] | 3.8M | – | – | 33.1 | – | – | 30.7 | 26.3 |
| SDL (M=7) [3] | 33.6M | 24.2 | 41.3 | 30.5 | 18.8 | 53.4 | 27.8 | 25.9 |
| DualCoOp (ours) | **0.02M** | **37.3** | **46.2** | **41.3** | **28.7** | **59.3** | **38.7** | **43.6** |
| *Generalized Zero-Shot Learning (GZSL)* | | | | | | | | |
| CONSE [45] | - | 11.5 | 5.1 | 7.0 | 9.6 | 7.1 | 8.1 | 2.1 |
| LabelEM [1] | - | 15.5 | 6.8 | 9.5 | 13.4 | 9.8 | 11.3 | 2.2 |
| Fast0Tag [69] | 0.61M | 18.8 | 8.3 | 11.5 | 15.9 | 11.7 | 13.5 | 3.7 |
| One Attention per Label [26] | ≥ 12.8M | 17.9 | 7.9 | 10.9 | 15.6 | 11.5 | 13.2 | 3.7 |
| LESA (M=10) [22] | ≥ 0.45M | 23.6 | 10.4 | 14.4 | 19.8 | 14.6 | 16.8 | 5.6 |
| BiAM [44] | 3.8M | – | – | 16.1 | – | – | 19.0 | 9.3 |
| SDL (M=7) [3] | 33.6M | 27.7 | 13.9 | 18.5 | 23.0 | 19.3 | 21.0 | **12.1** |
| DualCoOp (ours) | **0.02M** | **31.9** | 13.9 | **19.4** | **26.2** | 19.1 | **22.1** | 12.0 |

methods with both zero-shot setting (test only on unseen classes) and generalized zero-shot setting (test on both seen and unseen classes).

**Implementation.** We adopt ResNet-50 [19] similar to [3] as the visual encoder in DualCoOp for input resolution 224. Instead of learning class-specific prompts, we learn the class-agnostic context vectors with 64 context tokens ($N = 64$) for all classes, which is the only learnable part in DualCoOp. We optimize context vectors for 50 epochs with a batch-size 32/192 for MS-COCO/NUS-WIDE, respectively. During inference, we combine the learnt pair of context vectors with the class name for each class (either base class or novel class) and compute the text features. Other implementation details are the same as in Sec. 4.1

**Baselines.** To evaluate the effectiveness of DualCoOp in the zero-shot setting, we compare with the following baselines: (1). CONSE [45] adopts an ensemble of classifiers for unseen classes. (2). LabelEM [1] learns a joint image-label embedding. (3). Fast0Tag [69] and SDL [3] estimate one or multiple diverse principal directions of the input images. (4). Deep0Tag [49] and LESA [22] estimate the relevant regions via region proposals and attention techniques respectively. (5). BiAM [44] enhances the region-based features to minimize inter-class feature entanglement.

**Results.** Table 4-5 shows the comparison between DualCoOp and all SOTA methods of zero-shot learning and generalized zero-shot learning on MS-COCO and NUS-WIDE datasets. DualCoOp achieves the best F1 score in all cases with a very light learnable overhead (0.02M) and improves the performance of zero-shot learning (unseen labels) with a significant margin: F1 score improves by 12.5 @Top-3 on MS-COCO, and by 10.8 @Top-3 and 10.9 @Top-5 on NUS-WIDE. This shows the power of exploiting the pretrained alignment of textual and visual spaces in CLIP via DualCoOp to solve multi-label recognition.

### 4.3 Ablation Studies

**Effectiveness of Text Supervision.** To show the effectiveness of text supervision from label space, we compare the model learned with discrete label space ("Discrete Label") with four methods (SST [8], SARB [46], CoOp [71] and DualCoOp) which introduce the textual space to utilize the contextual correlation of labels in Table 6. We find that methods with text supervision usually perform better

Table 6: **Comparison among methods on MS-COCO using partial labels with the same initialization. All methods use parameters pretrained by CLIP [47].**

| Method | Text Supervision | 10% | 30% | 50% | 70% | 90% |
|---|---|---|---|---|---|---|
| Discrete Label | ✗ | 70.6 | 75.1 | 76.5 | 77.3 | 78.0 |
| SST | ✓ | 69.1 | 79.3 | 80.1 | 81.1 | 80.7 |
| SARB | ✓ | 75.5 | 79.0 | 80.4 | 80.8 | 80.8 |
| CoOp | ✓ | 63.0 | 68.5 | 69.2 | 71.5 | 75.0 |
| DualCoOp | ✓ | **78.4** | **81.0** | **82.0** | **82.5** | **82.8** |

Table 7: **Ablation on Linguistic Inputs for Zero-Shot Learning of MS-COCO.**

| | Linguistic Input | #P | ZSL | | | GZSL | | |
|---|---|---|---|---|---|---|---|---|
| | | | P | R | F1 | P | R | F1 |
| M0 | Contextless Classname | 0 | 5.2 | 12.9 | 7.4 | 3.5 | 4.1 | 3.8 |
| M1 | Hand-crafted Pos./Neg. Templates + Classname | 0 | 25.6 | 63.6 | 36.5 | 31.0 | 36.2 | 33.4 |
| M2 | Pos. Learnable Prompt + Classname ($N$=64) | 0.01M | 31.2 | 77.5 | 44.5 | 55.7 | 65.0 | 60.0 |
| M3 | Neg. Learnable Prompt + Classname ($N$=64) | 0.01M | 9.3 | 23.0 | 13.2 | 2.6 | 3.0 | 2.8 |
| M4 | Dual Learnable Prompts + Classname ($N$=64) | 0.02M | 35.3 | 87.6 | 50.3 | **58.4** | **68.1** | **62.9** |
| M5 | Dual Learnable Prompts + Classname ($N$=32) | 0.01M | **35.8** | **88.9** | **51.0** | 57.4 | 67.0 | 61.9 |

than the method only using discrete labels. However, when the semantic annotations are limited, text supervision sometimes yields worse performance (e.g. mAP of SST is $1.5\%$ lower than Discrete Labels with only $10\%$ of labels). CoOp [71] utilizes the visual-textual alignment. However, with the original multi-head attention and single positive prompt, it yields worse performance than Discrete Labels. To better utilize the well-pretrained alignment, DualCoOp learns dual context pairs and adopts class-specific region feature aggregation, which leads to great performance (*e.g.* $7.8\%$ higher than Discrete Labels with $10\%$ of labels) and quickly adapts to the dataset even with limited labels.

**Ablation of Prompt Design.** We compare our proposed dual learnable prompts with two hand-crafted prompts and one prompt learning method on the MS-COCO dataset with the zero-shot setting (see Table 7). Hand-crafted prompts can use either contextless class names [31] or manually designed prompt templates. In our experiments, we carefully choose the positive and negative prompt templates as "a photo of a [classname]" and "a photo without a [classname]". In contrast with performing the binary classification for each class with dual learnable prompts as the input, we also experiment with learning a single prompt of positive or negative contexts and use a chosen threshold (0.5 in our experiment) to make the prediction for each class. As we can see, the single positive prompt learning method (M2) performs better than non-learnable methods (M0 and M1), and a single negative learnable prompt (M3) achieves much worse accuracy than its positive counterpart (M2). However, when we include both positive and negative prompts, dual prompts (M4) performs even better than a single prompt, which indicates that DualCoOp learns complementary and beneficial information in the dual prompt pair. M4 also outperforms M2 in terms of mAP with both ZSL ($77.53\%$ vs. $64.74\%$) and GZSL ($69.8\%$ vs. $63.1\%$). To keep the same amount of learnable parameters as in single prompt settings, we also halve the token size (M5), and find that DualCoOp still outperforms two single prompts in M2 and M3 by large gaps, demonstrating the effectiveness of our dual-prompt design.

**Multi-Headed Attention vs. Class-Specific Region Aggregation.** In Table 8, we compare the adaptive ability of these two visual aggregation methods when training/testing with a larger resolution (see Table 8), which is crucial in multi-label recognition as spatial details matter. For a fair comparison, we only replace the class-specific region aggregation in DualCoOp with the original multi-headed attention layer in CLIP [48] at the end of the visual encoder. We adaptively resize the input feature map to match the input dimension of the multi-headed attention layer.

Table 8: **Comparison between multi-headed attention and class-specific feature aggregation on MS-COCO**

| Visual Aggregation | Finetune. | Train Res. | Test Res. | 10% | 30% | 50% | 70% | 90% |
|---|---|---|---|---|---|---|---|---|
| Multi-Headed Attention | ✗ | 224 | 224 | 70.4 | 74.1 | 74.8 | 75.4 | 75.7 |
| | ✗ | 224 | 448 | 65.9 | 70.2 | 71.2 | 72.0 | 72.1 |
| | ✗ | 448 | 448 | 72.1 | 75.5 | 76.5 | 77.1 | 77.3 |
| | ✓ | 448 | 448 | 74.1 | 77.6 | 78.2 | 78.5 | 78.4 |
| Class-Specific Feature Aggregation (DualCoOp) | ✗ | 224 | 224 | 73.1 | 76.4 | 77.7 | 78.2 | 78.4 |
| | ✗ | 224 | 448 | 76.0 | 78.1 | 79.5 | 80.3 | 80.5 |
| | ✗ | 448 | 448 | 78.4 | 81.1 | 82.0 | 82.5 | 82.8 |
| | ✓(Aggre. Func.) | 448 | 448 | **79.4** | **82.2** | **83.1** | **83.7** | 84.0 |
| | ✓(Image Enc.) | 448 | 448 | 57.8 | 69.0 | 73.7 | 80.9 | **84.2** |

As shown in Table 8, multi-headed attention is bonded to the pre-training image resolution (224 in CLIP), while our class-specific region aggregation benefits from the increased input resolution either during training or in inference. Our class-specific feature aggregation uses original weights, but actually performs better than finetuning the original multi-headed attention layer.

**Ablation of Finetuning DualCoOp.** To better exam-ine the effectiveness of finetuning for `DualCoOp`, we have implemented two different finetuning mechanisms for `DualCoOp`: (1). only finetune the weights in Class-Specific Region Feature Aggregation, (2). finetune all weights in the CLIP image-encoder. As shown in Table 8, finetuning aggregation function can stably improve the performance over the non-finetuning setting with differ-ent amounts of labels available. However, finetuning the entire visual encoder does not always outperform the non-finetuning baseline, especially under limited annotations ($\leq 70\%$ labels).

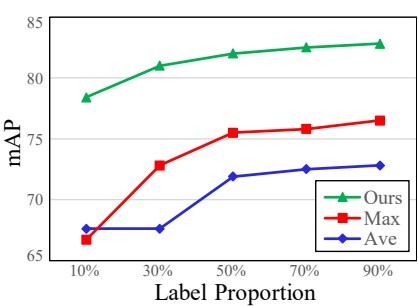

Figure 3: Comparison among different ag-gregations on MS-COCO using partial labels.

**Ablation of Aggregation Function.** We experiment with different functions to aggregate the regional logits for each class in Fig. 3. We compute final logits in three ways: (1) taking the average of logits at all spatial locations ("Ave"), (2) taking the region with the largest positive logit ("Max"), and (3) generating aggregating weights for all spatial locations via a softmax function over the positive logits ("Ours"). "Max" performs better than "Ave", which indicates the regional feature is more informative than the global feature in multi-label recognition. Furthermore, by taking account of both the regional and the global features, "Ours" gives the best performance.

## 5   Conclusion

In this paper, we propose a unified framework, `DualCoOp`, for two types of multi-label recognition with limited annotations: partial-label and zero-shot. `DualCoOp` utilizes powerful vision-language pretraining obtained from a web-scale dataset. By introducing a lightweight learnable overhead, it can quickly adapt to solve multi-label recognition after receiving a small amount of labels. In `DualCoOp`, we learn a pair of positive and negative prompts followed by the target class name as the linguistic input. Furthermore, to better aggregate visual region features for each class, we reformulate the original visual attention in the pretraining model as a class-specific region feature aggregation. We conduct extensive experiments for both partial-label MLR and Zero-Shot MLR across MS-COCO, VOC2007, and NUS-WIDE datasets, showing the efficacy of our proposed approach over state-of-the-art methods.

**Limitations.** Since the vision-language pretraining adopts a large Transformer-based language model and all labels need to be feed-forward through the text encoder, the large language model limits the size of the label set. Also, compared to training the model with both seen and unseen labels, we still get worse performance for the zero-shot unseen classes even though we have used 400M auxiliary samples in the pretraining. This highlights the difficulty of zero-shot MLR.

**Negative Societal Impacts.** Negative impacts of our research are difficult to predict, however, they are likely to be the usual pitfalls associated with deep learning models. These include susceptibility to adversarial attacks and data poisoning, dataset bias, and lack of interpretability. Other risks associated with the deployment of computer vision systems include privacy violations when images are captured without consent, or used to track individuals for profit, or increased automation resulting in job losses. While we believe that these issues should be mitigated, they are beyond the scope of this paper. Furthermore, we should be cautious of potential failures of the approach which could impact the performance/user experience of any high-level AI systems based on our research.

## Acknowledgement

This work is supported by NSF and DARPA LwLL. It reflects the opinions and conclusions of its authors, but not necessarily the funding agents.

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
