# DualCoOp: Fast Adaptation to Multi-Label Recognition with Limited Annotations (Supplementary Material)

**Ximeng Sun**[1]    **Ping Hu**[1]    **Kate Saenko**[1,2]
[1]Boston University, [2]MIT-IBM Watson AI Lab, IBM Research
{sunxm, pinghu, saenko}@bu.edu

## A    Different Prompt Length

We have provided the comparison of the performance of `DualCoOp` with different lengths of prompt context (i.e. $N = 2, 4, 6, 8, 16, 32, 64$) in all three different experiment scenarios (see Fig. 1 and 2). In MLR with partial labels, we learn class-specific prompts and thus `DualCoOp` performs good when $N$ is small, such as 8, 16. For zero-shot learning in MLR, we learn uniform prompts shared by all classes and it requires larger $N$ (e.g. 32 or 64) for good performance. In the main paper, we use $N = 16$ for all experiments of MLR with partial labels and use $N = 32$ for experiments in zero-shot learning.

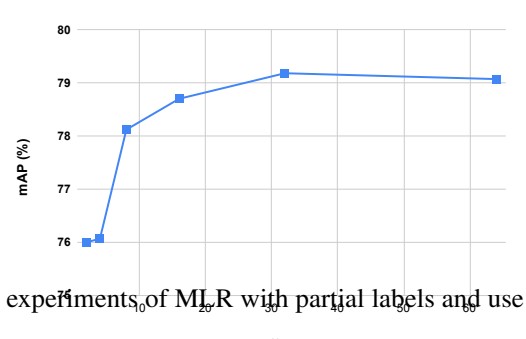

Figure 1: **MLR with Partial Labels at Different Prompt Length on MS-COCO [3]**

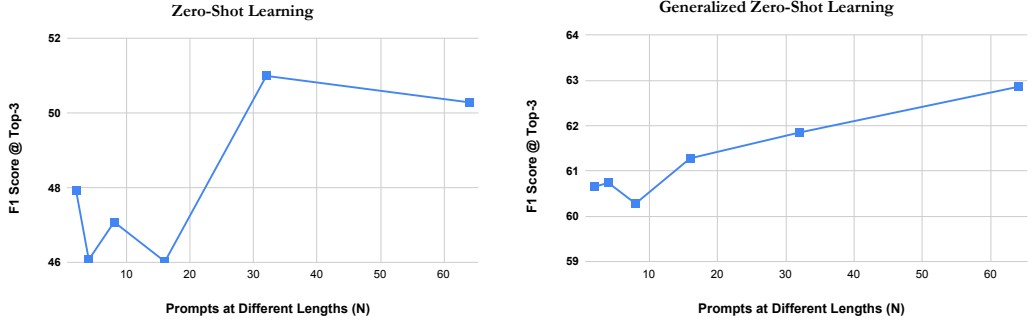

Figure 2: **Zero-Shot MLR with Different Prompt Length on MS-COCO [3]**

In the main paper, we set $N_+ = N_-$ for simplicity. Here, we conduct experiments in both partial-label MLC and Zero-Shot MLC settings to check the performance of different $N_-$s by controlling the $N_+$ as the same. As shown Table 1 and 2, F1-Score generally improves with larger $N_-$ in both partial label and zero-shot settings.

36th Conference on Neural Information Processing Systems (NeurIPS 2022).

Table 1: **Performance of different $N-$s with 10% labels on MS-COCO**

| $(N_+, N_-)$ | CP | CR | CF1 | OP | OR | OF1 | mAP |
|---|---|---|---|---|---|---|---|
| (16, 2) | 67.1 | 77.9 | 71.8 | 69.8 | 82.2 | 75.5 | 78.7 |
| (16, 4) | 67.7 | 77.6 | 72.1 | 70.3 | 81.8 | 75.6 | 78.7 |
| (16, 8) | 68.4 | 77.8 | 72.6 | 70.9 | 81.8 | 76.0 | 78.9 |
| (16, 16) | 69.1 | 77.5 | 72.6 | 71.4 | 81.6 | 76.2 | 78.7 |

Table 2: **Zero-Shot performance of different $N-$s on MS-COCO**

| $(N_+, N_-)$ | ZS-P | ZS-R | ZS-F1 | GZS-P | GZS-R | GZS-F1 |
|---|---|---|---|---|---|---|
| (32, 2) | 31.2 | 77.4 | 44.4 | 55.1 | 64.3 | 59.3 |
| (32, 4) | 33.1 | 82.1 | 47.1 | 57.1 | 66.6 | 61.5 |
| (32, 8) | 34.0 | 84.4 | 48.4 | 57.6 | 67.2 | 62.0 |
| (32, 16) | 34.8 | 86.6 | 49.7 | 57.5 | 67.1 | 61.9 |
| (32, 32) | 35.8 | 88.9 | 51.0 | 57.4 | 67.0 | 61.9 |

## B  Full performance of MLR with Partial Labels

In this section, we provide the average per-class and average overall precisions (CP and OP), recalls (CR and oR) and F1 scores (CF1 and OF1) of `DualCoOp` in the experiment of MLR with Partial Labels on MS-COCO [3], VOC2007 [2] and BigEarth [1] (see Table 3, 4 and 5 in supplementary material) as a supplementary for Table **??** and **??** in the main paper.

## C  Visualization of Class-Specific Region Feature Aggregation

We have visualized the class-specific region feature aggregation on MS-COCO dataset (in Fig. 3). We can see `DualCoOp` generates the high attention score at the correct objects.

Table 3: **Performance of MLR with partial labels on MS-COCO**

| Amount of Labels | CP | CR | CF1 | OP | OR | OF1 | mAP |
|---|---|---|---|---|---|---|---|
| 10% | 69.1 | 77.5 | 72.6 | 71.4 | 81.6 | 76.2 | 78.7 |
| 20% | 70.1 | 79.4 | 74.2 | 72.1 | 83.0 | 77.2 | 80.9 |
| 30% | 71.2. | 80.1 | 75.1. | 72.9. | 83.5 | 77.8 | 81.7 |
| 40% | 71.3 | 80.2 | 75.2 | 73.2 | 83.8 | 78.1 | 82.0 |
| 50% | 72.1 | 80.4 | 75.8 | 73.7 | 83.9 | 78.5. | 82.5 |
| 60% | 72.4 | 80.6 | 76.0 | 73.9 | 84.0 | 78.6 | 82.7 |
| 70% | 72.5 | 80.5 | 76.1 | 74.1 | 83.9 | 78.7 | 82.8 |
| 80% | 72.9 | 80.7 | 76.3 | 74.3 | 84.1 | 78.9 | 83.0 |
| 90% | 72.9 | 80.7 | 76.4 | 74.5 | 84.1 | 79.0 | 83.1 |
| 100% (No Finetune) | 73.2 | 80.8 | 76.6 | 74.6 | 84.2 | 79.1 | 83.2 |
| 100% (Finetune Aggre. Func.) | 75.7 | 80.4 | 77.8 | 77.1 | 83.7 | 80.3 | 84.2 |
| 100% (Finetune Img. Enc.) | 92.5 | 68.0 | 77.3 | 93.5 | 70.8 | 80.6 | 85.3 |

Table 4: **Performance of MLR with partial labels on VOC2007**

| Amount of Labels | CP | CR | CF1 | OP | OR | OF1 | mAP |
|---|---|---|---|---|---|---|---|
| 10% | 69.6 | 91.3 | 78.0 | 72.4 | 92.4 | 81.2 | 90.3 |
| 20% | 74.2 | 92.6 | 81.7 | 76.2 | 93.6 | 84.0 | 92.2 |
| 30% | 74.9 | 92.8 | 82.3 | 78.6 | 93.3 | 85.3 | 92.8 |
| 40% | 78.4 | 92.5 | 84.5 | 80.8 | 93.3 | 86.6 | 93.3 |
| 50% | 80.6 | 93.4 | 86.3 | 82.4 | 94.0 | 87.8 | 93.6 |
| 60% | 80.1 | 93.7 | 86.0 | 81.4 | 94.4 | 87.4 | 93.9 |
| 70% | 80.9 | 93.4 | 86.5 | 82.7 | 94.0 | 88.0 | 94.0 |
| 80% | 80.8 | 93.8 | 86.5 | 82.9 | 94.2 | 88.2 | 94.1 |
| 90% | 80.5 | 93.9 | 86.3 | 82.4 | 94.4 | 88.0 | 94.2 |
| 100% (No Finetune) | 81.2 | 94.1 | 86.8 | 83.2 | 94.5 | 88.5 | 94.4 |

Table 5: **Performance of MLR with partial labels on BigEartn**

| Amount of Labels | CP | CR | CF1 | OP | OR | OF1 | mAP |
|---|---|---|---|---|---|---|---|
| 10% | 76.9 | 84.3 | 78.8 | 71.9 | 85.9 | 78.3 | 88.2 |
| 20% | 81.6 | 94.2 | 86.9 | 73.4 | 93.1 | 82.1 | 92.9 |
| 30% | 83.7 | 93.1 | 87.4 | 75.7 | 92.5 | 83.3 | 93.1 |
| 40% | 82.7 | 93.9 | 87.2 | 75.8 | 92.0 | 83.1 | 93.5 |
| 50% | 81.3 | 93.2 | 85.9 | 74.4 | 90.4 | 81.6 | 93.7 |
| 60% | 86.2 | 92.3 | 88.9 | 80.2 | 91.1 | 85.3 | 94.3 |
| 70% | 86.0 | 92.8 | 88.8 | 79.4 | 91.7 | 85.1 | 94.2 |
| 80% | 85.1 | 94.8 | 89.2 | 77.9 | 93.2 | 84.9 | 94.1 |
| 90% | 83.9 | 94.4 | 88.2 | 77.2 | 93.4 | 84.5 | 94.7 |
| 100% (No Finetune) | 85.8 | 95.5 | 90.0 | 78.7 | 93.8 | 85.6 | 95.2 |

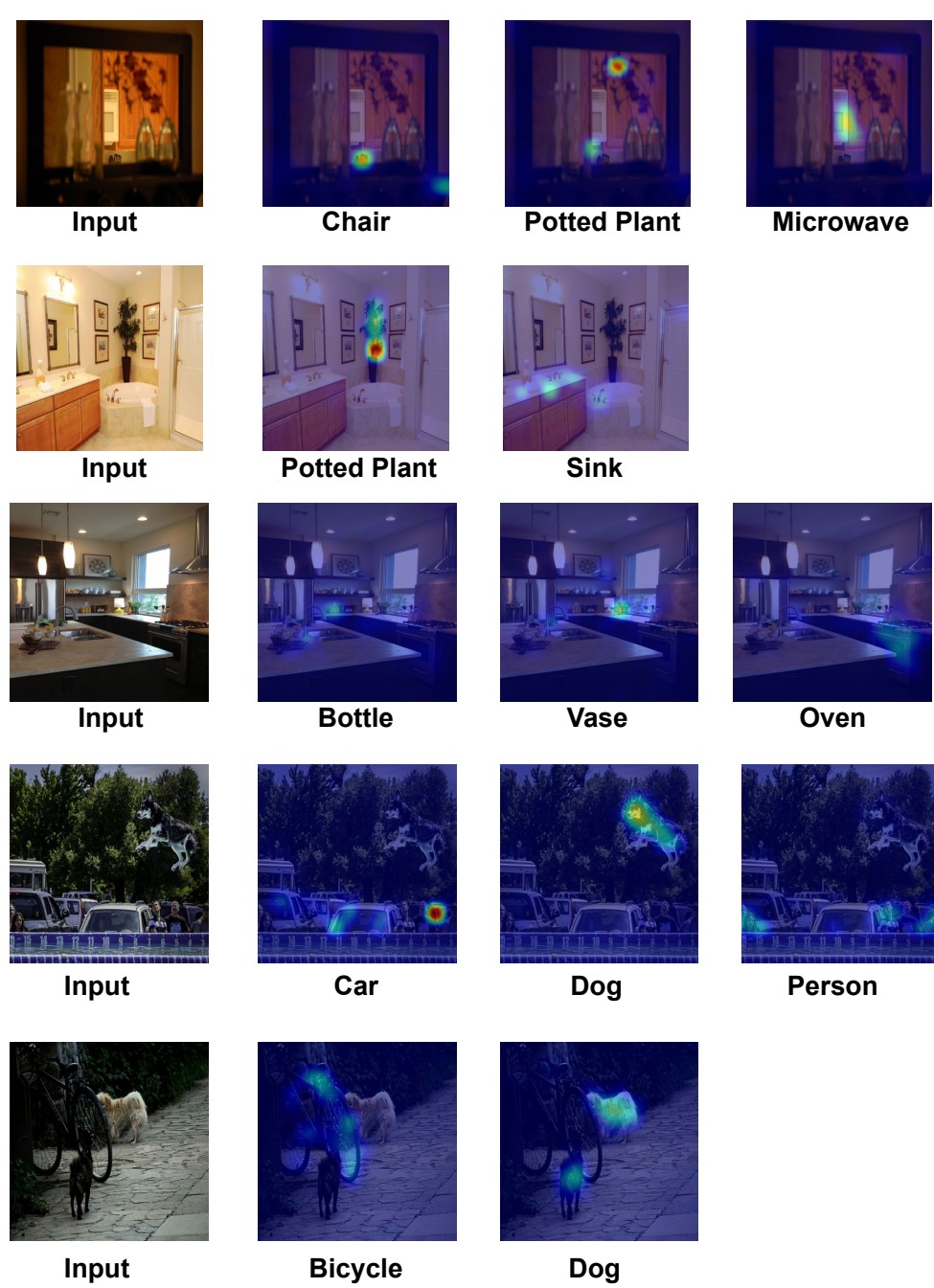

Figure 3: **Visualization of Class-Specific Region Feature Aggregation**