# OpenReview forum: "DualCoOp: Fast Adaptation to Multi-Label Recognition with Limited Annotations"
_NeurIPS.cc/2022/Conference — NeurIPS 2022 Accept_

### Official Review · Reviewer_3Z6j · 2022-07-10

**Rating:** 6
**Confidence:** 4
**Soundness:** 3 good
**Presentation:** 4 excellent
**Contribution:** 3 good

**Summary:**

The paper tackles the multi-label recognition (MLR) task in the low-label regime. The authors propose DualCoOp in order to adapt powerful vision-language models (such as CLIP) to solve MLR, by using dual (positive and negative) prompts. The paper presents results on partial label MLR and zero-shot MLR on datasets such as MSCOCO, PASCAL VOC etc.

**Questions:**

Would fine-tunning CLIP in conjunction with DualCoOp produce better results?

Would the method work on a dataset that is from a different domain as compared to what was CLIP trained on?

How would the results of training with 100% of data compare against some fully supervised methods?

How would using different amount of N+ and N- affect the performance? Would it be more useful to have more N+ than N-?

**Limitations:**

The authors discussed the limitations and the potential negative societal impact.

**Strengths And Weaknesses:**

The paper is well-written and it has a strong related work section. The experimental comparison with other state of the arts methods is very good the authors compare their method with the most recent works.
One main advantage of the method is that it only trains a light-weight head in order to adapt CLIP for MLR, which translates in having just a few learnable parameters.

The proposed method is similar to CoOP, so a more in-depth comparison would be beneficial.

Some ablations of the Class-Specific Region Feature Aggregation module are missing.

Overall the presented work is well-written and is easy to understand the overall idea. Also the method achieves very good results as compared to other works.

---

> ### Author Response · Authors · 2022-08-02
> **Response to Reviewer 3Z6j**
>
> Thanks for raising this important question! To better examine the effectiveness of finetuning for DualCoOp, we have implemented two different finetuning mechanisms for DualCoOp: (1). only finetune the weights in Class-Specific Region Feature Aggregation, (2). finetune all weights in the CLIP image-encoder.
> As shown in the following table, finetuning aggregation function can stably improve the performance over the non-finetuning setting with different amounts of labels available. However, finetuning the entire visual encoder does not always outperform the non-finetuning baseline, especially under limited annotations (e.g. $\leq$70\% labels).
>
> |  |  | |   | | |
> |  ----  | ----  | ----  | ----  | ----  | ----  |
> |  Method \ Ratio   | 10\%  | 30\%  | 50\%  | 70\%  | 90\%  |
> |No Finetune | 78.7 | 81.0 | 82.0 | 82.5  | 82.8  |
> |Finetune Aggre. Func. | 79.4 | 82.2 | 83.1 | 83.7 | 84.0|
> |Finetune Visual-encoder | 57.8  | 69.0 | 73.7 | 80.9 |  84.2|
> |  |  | |   | | |
>
> **Q2: Would the method work on a dataset that is from a different domain as compared to what was CLIP trained on?**
>
> Indeed it is interesting to experiment with datasets that are out of the domain of CLIP pretraining data. Yet, CLIP pretraining data is not publicly available and it is plausible that CLIP pretraining data covers many coarse and fine-grained visual domains since it performs well in the zero-shot evaluation for many downstream tasks. Instead, we experiment on a Remote Sensing Image dataset BigEarth, whose domain is far from the domains of the datasets in the mainstream papers (i.e. PASCAL VOC, MS-COCO, and NUS-WIDE) ,and hopefully BigEarth is out of the domain of CLIP pretraining data. We compare DualCoOp with SARB* (the best baseline in Table 1 of the main paper). Both DualCoOp and SARB* are initialized with CLIP pretrained weights. As shown in the following table, DualCoOp consistently outperforms SARB* on BigEarth as well as the datasets in the main paper.
>
> |  |  | |  | | |
> |  ----  | ----  | ----  | ----  | ----  | ----  |
> |  Method \ Ratio   | 10\%  | 30\%  | 50\%  | 70\%  | 90\%  |
> |SARB* | 77.2 | 83.6 | 87.4 | 91.6  | 93.5  |
> |DualCoOp | 88.2 | 93.1 | 93.7 | 94.2 | 94.7|
> |  |  | |   | | |
>
> **Q3: How would the results of training with 100\% of data compare against some fully supervised methods?**
>
> Thanks for this comment. We found that on MSCOCO when using 100\% of training labels and finetuning the visual encoder, our method achieves  85.2\% mAP, outperforming previous SOTA approaches like ASL (85.0\% mAP) and CSRA (83.5\% mAP) with the same Res101 backbone.
> Without finetuning the visual encoder, we notice that our method achieves 83.2\%  mAP on MSCOCO with 100\% labels, which is still comparable to the performance of ASL and CSRA.
> We hypothesize that the performance drop of non-finetuning setting is due to the noise in text-image pairs for pre-training the CLIP model. This also shows that it is non-trivial to address MLR task with the pretrained CLIP model, though it is pre-trained with larger-scale data. We will update our paper with results under 100\% annotations.
>
> **Q4: How would using different amount of N+ and N- affect the performance? Would it be more useful to have more N+ than N-?**
>
> In the main paper, we set $N_+ = N_-$ for simplicity. Here, we conduct experiments in both partial-label MLC and Zero-Shot MLC settings to check the performance of different $N_-$s by controlling the $N_+$ as the same. As shown in the following tables, F1-Score generally improves with larger $N_{-}$  in both partial label and zero-shot settings.
>
>
> |  | |  | |  |    |   |      |
> |             ----          |      ----     |      ----     |      ----     |      ----     |      ----     |      ----     |      ----     |
> |  ($N_+$, $N_-$)&nbsp; |      CP   |      CR    |      CF1   |      OP    |      OR    |      OF1    |      mAP    |
> |(16, 2) | 67.1 | 77.9| 71.8 | 69.8 | 82.2 |	75.5 | 78.7 |
> |(16, 4) | 67.7 | 77.6 | 72.1 | 70.3 | 81.8| 75.6 | 78.7 |
> |(16, 8) | 68.4 | 77.8 | 72.6 | 70.9 |  81.8 | 76.0 | 78.9 |
> |(16, 16) | 69.1 | 77.5 | 72.6 |	71.4 | 81.6 | 76.2 | 78.7 |
> |  | |  | |  |    |   |      |
>
> |  | |  | |  |    |   |
> |             ----          |      ----     |      ----     |      ----     |      ----     |      ----     |      ----     |
> |  ($N_+$, $N_-$)&nbsp; |      ZS-P    |      ZS-R  |    ZS-F1  |      GZS-P  |     GZS-R   |      GZS-F1    |
> |(32, 2) | 31.2 | 77.4 |	44.4 | 55.1	| 64.3 | 59.3 |
> |(32, 4) | 33.1 | 82.1 |	47.1 | 57.1 | 66.6 | 61.5 |
> |(32, 8) | 34.0 |  84.4 | 48.4 |  57.6 | 67.2 |	62.0 |
> |(32, 16) | 34.8 | 86.6 | 49.7 |  57.5 |	67.1 | 61.9 |
> |(32, 32) | 35.8 | 88.9 | 51.0 |	57.4 | 67.0 | 61.9 |
> |  | |  | |  |    |   |

---

> > ### Comment · Reviewer_3Z6j · 2022-08-07
> > **Rebuttal feedback**
> >
> > Thank you for your rebuttal and for the extra experiments! The rebuttal answered all my questions!
> >
> >  I would suggest to include the extra experiments in the revised version of the paper (at least in the supp material). I would say that the results presented in Q3 and Q4 are very valuable!
> >
> > I will keep my original rating "Weak accept".

---

> > > ### Author Response · Authors · 2022-08-07
> > > **Response to Rebuttal feedback**
> > >
> > > Thank you for reading our rebuttal. We will update our paper with these experiments in the next version.

---

### Official Review · Reviewer_n9WS · 2022-07-10

**Rating:** 5
**Confidence:** 4
**Soundness:** 3 good
**Presentation:** 3 good
**Contribution:** 2 fair

**Summary:**

The paper proposes a method for the partial- and zero-label multi-label image classification problems. Specifically, the paper extends on CoOp [70], where a learnable prompt is used combined with the image to tune CLIP-like architecture to downstream tasks; and proposes to use two learnable prompts, one for the positive label (+1) and the other for the negative label (-1) of the same class, dubbed DualCoOp. Moreover, the paper also proposes to calculate the similarity scores between the feature from each image region and the text features, then aggregates all the logits. The proposed method is evaluated on VOC, COCO, and NUS-Wide in both the partial-label setting between 10-90% labels and the zero-label(shot) setting.

**Questions:**

Please refer to the weaknesses section for details.  I would lean more towards accepting if the authors could answer my questions and provide the data points during rebuttal.

**Strengths And Weaknesses:**

The main strengths of the paper are:

1.  The paper proposes a simple yet effective idea to extend CoOp [70] to DualCoOp, with two learnable prompts for negative and positive labels. This idea intuitively is suitable for multi-label learning and performs well based on the experimental results shown in the paper. However,  I do have some questions and doubts about the explanation of the effectiveness of the proposed method and ablation study.
2. The paper is well written and easy to follow. The proposed idea should be easily reproducible. Moreover, the experiments are quite thorough in general, albeit I would like to see more ablation of the baseline methods, which I will explain in the weaknesses and questions.

The main weaknesses of the paper are:

1. As the idea is based on CoOp [70], I would like to see a thorough ablation compared with CoOp, especially in Table 1 and for the partial-label setting. I understand that M3 in Table 8 is basically the same as CoOp, but I think the threshold choice of 0.5 is not fair to methods that rely on thresholds. I would like to see the mAP number of Table 8, which does not rely on threshold selection.
2. The paper overempharized the idea of learnable parameters and ignored what will this actually translates into real-world effect. Based on my understanding, although the learnable parameters are small, the actual inference latency would be greater than a straightforward multi-label classification network. I would like to see actually training/inference time and memory consumption in the same hardware environment rather than just the parameters.
3. Missing info on other baseline methods and possible unfair comparisons. I appreciate the papers' efforts in using CLIP initialization for  SST [7] and SARB [45], but for researchers that are not absolutely certain about the details of the compared method, I would suggest listing the training/inference resolution, backbone architecture, and pre-training data for each method. Pre-training on 400M obviously would bring benefits to downstream tasks than just IN-1K (as shown in the comparison in SST and SARB), so I would say most of the comparisons in the paper do not have much value in understanding which method is better in what situation, besides adding one more datapoint confirming CLIP is good. I would rather see more comparisons with CLIP-based methods, such as CoOp.

---

> ### Author Response · Authors · 2022-08-02
> **Response to Reviewer n9WS**
>
> **Q1: I would like to see a thorough ablation compared with CoOp, especially in Table 1 and for the partial-label setting. I would like to see the mAP number of Table 8, which does not rely on threshold selection.**
>
> Thanks for this valuable suggestion. We guess that by *M3 in Table 8*,  reviewer n9WS  may be referring to  *M2 in Table 5* of the paper. We want to note that M2 in Tab.5 differs a bit from CoOp as M2 is based on our proposed Class-Specific Region Feature Aggregation, while CoOp utilizes the original multi-head attention. In Tab.6 of the paper, we have shown the better effectiveness of Class-Specific Region Feature Aggregation as compared to the multi-head attention.
> As requested by R3, we also directly compare CoOp and DualCoOp for partial label multi-label recognition on MS-COCO in the following table.  As we can see, DualCoOp consistently outperforms CoOp with different amounts of labels.
>
> |  |  | |   | | |
> |  ----  | ----  | ----  | ----  | ----  | ----  |
> |  Method \ Ratio   | 10\%  | 30\%  | 50\%  | 70\%  | 90\%  |
> | CoOp  |  63.0 |  68.5 |  69.2 |  71.5 |  75.0 |
> | DualCoOp   | 78.7 | 81.0 | 82.0 | 82.5  | 82.8 |
> |  |  | |   | | |
>
> We also evaluate mAP for M2 and our full model in both the zero-shot setting and the generalized zero-shot setting. In the zero-shot setting, M2 achieves mAP 64.74\% while DualCoOp (our full model) achieves mAP 77.53\%. In the generalized zero-shot setting, M2 achieves mAP 63.1\% while DualCoOp achieves mAP 69.8\%.
>
> We will incorporate more results into our paper.
>
> **Q2: I would like to see actually training/inference time and memory consumption in the same hardware environment rather than just the parameters.**
>
> Thanks for bringing this up. For the current multi-label recognition task, the categories are pre-set before inference, (i.e. we already know which class we would like to consider during inference.) In this case, we compute the text features for each class from the learned prompts and the class name ahead of the inference. Then we use the pre-computed text features to predict each image during the test. Since the text features are pre-computed (very light-weighted computing overhead), the text encoder is not executed during inference. The latency time and memory consumption of DualCoOp are the same as other baselines when using the same backbone for the image encoder. During training, CLIP-based methods slightly raise latency time and memory consumption since image and text encoders are both executed during the forward, and only prompts are updated in DualCoOp. In the following table, we list the comparison between DualCoOp and SARB in terms of latency time and memory consumption during the training and inference using the same device (Nividia A100 GPU).  We will update our paper with more  comparisons of run-time efficiency.
>
> |           |             |           |             |            |          |
> |     ----          |      ----          |      ----          |      ----          |      ----          |       ----          |
> | Method                   |    Training Latency&nbsp;&nbsp;&nbsp;      |      Training Memory&nbsp;&nbsp;&nbsp;       |  Testing Latency &nbsp;&nbsp;&nbsp;   |     Testing Memory&nbsp;&nbsp;&nbsp;     |   Avg. MAP |
> |                    |               (ms/mg)           |       (GB/mg)       |   (ms/mg)     |      (GB/mg)      |   （\%） |
> | SARB*        | 4.7        | 0.21        | 4.0        | 0.13        |     79.4          |
> | DualCoOp  | 5.3        | 0.22         | 4.0        | 0.06  |      81.9         |
> |           |             |           |             |            |          |
>
>
> **Q3: I would rather see more comparisons with CLIP-based methods, such as CoOp.**
>
> Thanks for the comment. As requested, we provide CoOp results for partial-label MLR in the table of Q1, which shows DualCoOp achieves better performance than CoOp in this setting. Also, we compare M2 (in Table 5) with DualCoOp for zero-shot setting, and DualCoOp gives better performance in Top-3 F1 and mAP in both zero-shot and generalized zero-shot settings. For baseline methods, we state in L204-205 that all baselines in Partial Label Multi-Label Recognition are using ResNet-101 and resolution 448x448 (the same statement is also made in [7] and [45]), which is the same setting adopted in DualCoOp in Table 1. For zero-shot settings, all baselines are using ResNet50 with input resolution 224x224, the same as DualCoOp. All baselines, except SARB* and SST* , are initialized with imagenet-pretrained weights. We agree CLIP 400M pretraining is powerful but we have shown that it is not trivial to apply it to multi-label classification (see the comparison between DualCoOp and CoOp).  DualCoOp outperforms both models with either imagenet initialization weights or CLIP initialization weights.

---

### Official Review · Reviewer_BkXD · 2022-07-12

**Rating:** 6
**Confidence:** 3
**Soundness:** 3 good
**Presentation:** 3 good
**Contribution:** 3 good

**Summary:**

The paper solves the multi-label recognition tasks with limited annotations via pre-trained vision-language models (e.g., CLIP). The approach involves the learning of dual prompts, which significantly reduces the amount of learnable parameters. The experiments were conducted on partial-label and zero-shot MLR benchmarks and showed state-of-the-art performance.


**Questions:**

1. Although this paper aims to address limited-annotation MLR, the proposed method should also work for typical (supervised) MLR, in which all labels are available for optimization. I would like to know authors' thoughts on this point.
2. I am not quite sure how the method works for zero-shot MLR. I can understand that a pair of positive and negative prompts for each class is learned when solving MLR with partial labels. In that case inference can be made by directly comparing the positive and negative logits to make a decision for each label. For zero-shot MLR, only a pair of prompts are learned (they are shared for all classes in zero-shot MLR). In that case, how does the inference proceed?


**Limitations:**

Yes, authors have provided two paragraphs in the conclusion section to describe the limitations and potential negative societal impact.

**Strengths And Weaknesses:**

This paper unifies partial-labeled and zero-shot multi-label recognition problems and presents a solution to address both challenging problems. The approach takes advantage of powerful vision-language models and learns a pair of positive and negative prompts as the textual input, which is different to previous MLR methods. Results are promising in that the proposed method outperformed state-of-the-arts in partial-labeled and zero-shot MLR benchmarks. Ablation studies on design choices are also provided.

---

> ### Author Response · Authors · 2022-08-02
> **Response to Reviewer BkXD**
>
> **Q1: Although this paper aims to address limited-annotation MLR, the proposed method should also work for typical (supervised) MLR, in which all labels are available for optimization. I would like to know authors' thoughts on this point.**
>
> Thanks for this comment. Indeed, our method does not only work for limited-label settings, but is also effective for full label training. On MS--COCO when using 100\% of training labels and finetuning the visual encoder, our method achieves  85.2\% mAP, outperforming previous SOTA approaches like ASL (85.0\% mAP) and CSRA (83.5\% mAP) with the same Res101 backbone.
> If not finetuning the visual encoder, we notice that the performance drops to 83.2\%  mAP, which is still comparable to the performance of ASL and CSRA.
> Such a performance drop may be due to the noise in the text-image pairs used for pre-training the CLIP model and the misalignment of the objectives between the pre-training task and MLR task. This also shows that it is non-trivial to address the MLR task with the pretrained CLIP model, even though it is pre-trained with larger-scale data.
> We will update our paper to add these results of training with full annotations.
>
>
> **Q2: For zero-shot MLR, only a pair of prompts are learned (they are shared for all classes in zero-shot MLR). In that case, how does the inference proceed?**
>
> Thanks for this comment. For Zero-shot Multi-Label Recognition, we learn a single pair of prompts for all labels, i.e. the prompts are label-agnostic. During  inference, we combine the pair of learned prompts with the class name for each class (either base class or novel class) and compute the text features. We will clarify this in the next version.

---

### Official Review · Reviewer_k3ET · 2022-07-12

**Rating:** 5
**Confidence:** 4
**Soundness:** 4 excellent
**Presentation:** 3 good
**Contribution:** 3 good

**Summary:**

This paper addresses the low-label regime of multi-label recognition by introducing an image-text framework. A light learnable overhead is added to pre-trained vision-language models which can quickly adapt to MLR tasks with unseen classes. On two selected benchmarks the proposed method demonstrated promising results.

**Questions:**

What are the key theoretical contributions of this paper?

What are the most challenging technical problems addressed in this paper?

When handle the positive and negative prompts, how is it compared to contrastive learning?

**Limitations:**

The theoretical contribution is limited despite a fancy task with a novel and practical framework design.

**Strengths And Weaknesses:**

The MLR task with a low-label regime is a valuable problem in practice.

Introducing prompts improves the feasibility of the solution.

Extensive experimental results.

Good presentation.

- Lack of theoretical analysis of the logits.

---

> ### Author Response · Authors · 2022-08-02
> **Response to Reviewer k3ET**
>
> **Q1: What are the key theoretical contributions of this paper?**
>
> Thank you for this valuable question. In this work, we make two modeling contributions.
>
> - First, we propose to model partial- and zero-shot multi-label recognition as a single, unified problem.
>
> - Second, we propose to explicitly model both a class's existence and non-existence through learning dual contexts. In contrast to existing models that only focus on predicting a class's existence, our method introduces a novel formulation for analyzing MLR. The effectiveness of this choice is validated experimentally.
>
>
>
>
> **Q2: What are the most challenging technical problems addressed in this paper?**
>
> Thank you for this important comment. In this work, we address two technical challenges.
> - How to exploit pretrained vision-language models like CLIP for Multi-Label Recognition (MLR) with Limited Annotations? Unlike single-label classification focusing on image-level representation, MLR requires the models to perceive spatially localized objects. To address this challenge, we propose the Class-Specific Region Feature Aggregation to highlight information from related regions and bypass unrelated areas. We demonstrate our method's effectiveness in Fig-3 and Tab-6 of the paper as well as  Fig-3 of the supplementary.
>
> - How to better exploit the rich semantic context learned by pretrained vision-language models like CLIP? Adopting a positive context to predict the existence of a target class is an intuitive and popular way, yet may pose challenges for deciding a proper classification threshold, especially under limited annotations. We resolve this challenge by explicitly learning both the positive context and the negative context, which work together to exploit complementary information from CLIP.
>
> **Q3: When handling the positive and negative prompts, how is it compared to contrastive learning?**
>
> Thanks for this insightful comment. In a high-level sense, our method is similar to contrastive learning, as it compares the predictions from the positive context and the negative context.  Yet, we want to note that DualCoop is different from contrastive learning in three ways from a methodology perspective.
>
> - Contrastive learning aims to learn a unified mapping function, while in DualCoop we learn the positive context and negative context to produce two separate encodings.
>
> - Contrastive learning uses pos/negative pairs, while DualCoOp works with single-image labels.
>
> - In contrastive learning, negative samples needs to be carefully and amply sampled from different categories to increase inter-class variance and decrease intra-variance of data in the embedding space, while in DualCoop we focus on exploiting both the positive and negative semantic context from the pretrained vision-language model.

---

### Author Response · Authors · 2022-08-02
**Thanks to the reviewers for their thoughtful feedback**

We thank the reviewers for their thoughtful feedback! To summarize the positive points, reviewers thought that

- Multi-Label Recognition (MLR) with low labels is a valuable problem in practice;

- Our idea to introduce learnable prompts to the MLR problem is simple and effective. It is intuitively suitable for MLR and only trains a light-weight head containing few learnable parameters;

- We have shown extensive experiments with promising results outperforming SOTA in partially-labeled and zero-shot MLR benchmarks, and provide ablation studies of our design choices;

- The paper is well-presented, easy to follow, and has a strong related work section;

- The proposed idea should be easily reproducible.

---

### Meta-Review · Area_Chair_QaGN · 2022-08-26

**Recommendation:** Accept
**Confidence:** Certain

**Metareview:**

This paper’s DualCoOp extends the previous CoOp prompt learning framework to multi-label and multi-label zero-shot recognition. Reviewers were broadly positive, appreciating the writing, good results, and overall idea of exploiting pre-trained CLIP models for MLR. Questions focused on comparison to vanilla CoOp, evaluation in the fully supervised regime, inference cost and impact of fine-tuning. These were all generally resolved during author feedback phase. Since all reviewers are positive and questions are resolved, I recommend accept.


**Award:**

No

---

### Decision · Program_Chairs · 2022-09-14

Accept